# Design, Synthesis, and Bioactive Screen In Vitro of Cyclohexyl (*E*)-4-(Hydroxyimino)-4-Phenylbutanoates and Their Ethers for Anti-Hepatitis B Virus Agents

**DOI:** 10.3390/molecules24112063

**Published:** 2019-05-30

**Authors:** Xinhua Cui, Min Zhou, Jie Tan, Zhuocai Wei, Wanxing Wei, Peng Luo, Cuiwu Lin

**Affiliations:** Department of Chemistry, Guangxi University, Nanning 530004, China; cuixinhua123@163.com (X.C.); leelmm@gxu.edu.cn (M.Z.); yulitanjie@163.com (J.T.); muse00@163.com (Z.W.); lcl40261461@163.com (P.L.); lincuiwu@gxu.edu.cn (C.L.)

**Keywords:** design, synthesis, anti-HBV activity, structure–activity relationship, molecular docking

## Abstract

A series of oxime Cyclohexyl (*E*)-4-(hydroxyimino)-4-phenylbutanoates and their ethers were designed, synthesized, and evaluated for anti-hepatitis B virus (HBV) activities with HepG 2.2.15 cell line in vitro. Most of these compounds possessed anti-HBV activities, and among them, compound 4B-2 showed significant inhibiting effects on the secretion of HBsAg (IC_50_ = 63.85 ± 6.26 μM, SI = 13.41) and HBeAg (IC_50_ = 49.39 ± 4.17 μM, SI = 17.34) comparing to lamivudine (3TC) in HBsAg (IC_50_ = 234.2 ± 17.17 μM, SI = 2.2) and HBeAg (IC_50_ = 249.9 ± 21.51 μM, SI = 2.07). Docking study of these compounds binding to a protein residue (PDB ID: 3OX8) from HLA-A2 that with the immunodominant HBcAg18–27 epitope (HLA-A2.1- restricted CTL epitope) active site was carried out by using molecular operation environment (MOE) software. Docking results showed that behaviors of these compounds binding to the active site in HLA-A protein residue partly coincided with their behaviors in vitro anti-HBV active screening.

## 1. Introduction

Hepatitis B virus (HBV) remains a serious health problem for leading risks of liver related diseases for many people in the world. An estimation of 257 million people is chronically infected with HBV in the world, and among them, about 15–40% develop liver cirrhosis, hepatic failure, and hepatocellular carcinoma. More than 887,000 people die every year for complications of HBV [1,2]. Therapies of hepatitis B infection currently include methods of direct-acting antivirals (just as with nucleoside analogs) and host-targeting antivirals (just as with interferon) [3]. The nucleoside analogs in treatment of anti-HBV play a role of Troy horse in synthesis of HBV DNA and suppressed replication of HBV, but they are not effective to eliminate virus completely from patients [4,5,6]. Meanwhile, therapies of HBV with nucleoside analogs in the long term cause side effects and resistance [7,8,9]. Therefore, it is still a tremendous challenge to develop new anti-HBV agents for further improvement of anti-HBV therapy. Many non-nucleoside analog compounds with anti-HBV activities have been found and invented from synthesized compounds [10,11] and natural products [12,13,14]. Researches showed that HLA-A2 with the immunodominant HBcAg18–27 epitope (HLA-A2.1- restricted CTL epitope) binding peptides of vaccine or of HBcAg initiated specific respond of T cell and resolved cute HBV infection [15,16,17,18,19]. Other peptides specifically bound to the HLA-A2 residue activate CTL responds to prevent infection and eliminate HBV [20,21]. Therefore, a proposal can be implied that other non-peptide compounds specifically interact with protein residue in HLA-A2.1 restricted CTL epitope may initiate CTL respond to prevent infection of HBV. A protein residue from HLA-A2.1-restricted CTL epitope (PDB ID: 3OX8) interacted intensively with anti-HBV active compounds in docking investigations, was used as a virtual target to design anti-HBV new compounds in our previous works. Series of oxime-contained compounds with C_6_-C_3_ and C_6_-C_4_ skeleton were designed, synthesized, and assayed for anti-HBV activities. Results showed these compounds possessed significant anti-HBV activities [22,23,24,25]. On the basis of these results, we designed and synthesized a series of another part of novel oximes and their derivatives with C_6_-C_4_ skeletons, and screened for anti-HBV activities in our present works. All these compounds were evaluated theoretically with a mimic target HLA-A protein (PDB ID 3OX8) by molecular docking using molecular operating environment (MOE) method. Results showed these compounds with significant anti-HBV activities in vitro partly coincided with molecular docking.

## 2. Results and Discussion

### 2.1. Chemistry

A series of oxime and oxime ether derivatives of (E)-4-(alkyl-imino)-4-(4-substitute-phenyl) cyclohexylbutanoate were designed for anti-HBV agents according molecular docking conclusions. General synthesis of these compounds was depicted in Scheme 1. Compounds **2A**–**2D** were prepared by Friedel-Crafts reaction of succinic anhydride with substituted benzene in yields of 95–99%. Compounds **3A**–**3D** were prepared by reaction of compounds **2A**–**2D** with cyclohexanol catalyzed with p-CH_3_C_6_H_4_SO_3_H (TsOH) in yields of 81-89%. Oxime ether derivatives (**4A-1**–**4A-3**, **4B-1**–**4B-3**, **4C-1**–**4C-3**, **4D-1**–**4D-3**) were obtained by reaction of substitute-hydroxylamine hydrochloride (hydroxylamine hydrochloride methoxylamine hydrochloride and benzylhydroxylamine hydrochloride) with compounds **3A**–**3D** in yields of 75–88%. The structures of the newly synthesized compounds were characterized by ^1^H NMR, ^13^C NMR and MS. ^1^H NMR spectra of the derivatives showed a singlet at about 8.69–9.23 ppm corresponding to N-O-H proton in oxime compounds **4A-1**, **4C-1** and **4D-1** (Compound **4B-1** was deuterated in CH_3_OD). The signal at 6.92–7.64 ppm of multiplets in the region attributed to aromatic hydrogens of the phenyl ring, especially 6.92–7.63 and 7.37–7.69 ppm of two doubles in the region attributed to the phenyl ring of R_1_. ^13^C NMR spectrum for the derivatives showed signals at 155.68–158.24, 171.90–172.35 corresponding to C=N and C=O, respectively.

### 2.2. Anti-HBV Activities *In Vitro*

Inhibition of secreting HBsAg and HBeAg of these synthesized compounds were assayed in HepG 2.2.15 cells with lamivudine (**3TC**) as a positive control. Results of cytotoxicity and anti-HBV activities were showed in Table 1 and Figure 1. Many compounds showed more effective in inhabiting secretion of HBsAg and HBeAg than that of lamivudine, and among them compound **4B-2** showed inhibition of HBsAg (IC_50_ = 63.85 μM, SI = 13.41) and HBeAg (IC_50_ = 49.39 μM, SI = 17.34), with low toxicity (CC_50_ = 856.2 μM), and compound **4B-3** in inhibiting secretion of HBsAg (IC_50_ = 78.46 μM,) and HBeAg (IC_50_ = 84.57 μM) and low toxicity (CC_50_ = 859.5 μM) with SI values (SI_HBsAg_ = 12.90, SI_HBeAg_ = 10.89). When R=H, all oximes in the four groups decreased inhibitions of HBsAg in 505.7, 257.4, 489.3, and 352.4μM, respectively, HBeAg in 375.0, 283.7, 404.7, and 373.4 μM, respectively, and elevated cytotoxicity than their relative ethers.

### 2.3. Molecular Docking Study

The docking software MOE 2008.10 was used for molecular docking of oxime derivatives and a protein residue in HLA-A2.1-restricted CTL epitope (PDB ID: **3OX8**) was selected for docking study. Molecular docking results clearly revealed interactions between ligands and protein in the active site. Affinity scoring function ΔG of the protein-ligand complexes and other docking results were shown in Table 2. Significant interactions with protein residue in docking were compound **4A-3** and **4D-3** for −22.8858 kcal/mol and −21.2040 kcal/mol, respectively. Two hydrogen bonds of lengths 2.29 and 2.58 Å for N of O-N in the oxime group with Tyr27 and Tyr63, respectively, was found for compound **4A-3** (Figure 2 and Figure 3).

### 2.4. Structure-Activity Relationship (SAR)

Oximes (**4A-1**, **4B-1**, **4C-1** and **4D-1**) which R=H showed weaker anti-HBsAg and anti-HBeAg activities, weaker affinities to the selected target protein in docking than those compounds which R=Me and benzyl. When these oximes were etherified, anti-HBV activities of their ethers increased, meanwhile cytotoxicity of these ethers decreased by comparing to their relative oximes. Methyl ethers **4A-2**, **4B-2**, **4C-2** and **4D-2** showed more significant anti-HBV active in their relative groups (Figure 4). Results also revealed that substitute groups in phenyl (groups **4B**, **4C** and **4D**) increased anti-HBV activities by comparing to relative compounds in **4A** group. Docking results showed that N or O atoms in ON fragment and O atom in C=O groups in the oxime derivatives interacted with ammonic acid residues by hydrogen bond, and their docking affinity partly coincided with their anti-HBV activities in vitro. Similar structure-activity relationships were found in our previous works [23].

## 3. Materials and Methods

Melting points were measured on an uncorrected WRX-4 electrothermal melting point apparatus (Shanghai, China). ^1^H NMR and ^13^C NMR spectra were recorded on Bruker AVANCE III HD600 (^1^H/^13^C, 600MHz/150MHz) spectrometer (Bruker, Bremerhaven, Germany) using TMS as an internal standard. The mass spectra were recorded on a Finnigan LCQ Deca XP MAX mass spectrometer (Thermo Fisher, San Jose, CA, USA) equipped with an ESI source and an ion trap analyzer in the positive ion mode/in the negative ion. Silica gel GF-254 was used in thin-layer chromatography, and silica gel H used in flash column chromatography (Qingdao Haiyang Chemical, Qingdao, China). All solvents and reagents were analytical grade. The purity of target compounds was assessed on the basis of analytical HPLC (Thermo Fisher Ultimate 3000), and the results were > 95%.

### 3.1. Chemistry

#### 3.1.1. Preparation of 4- oxo-4-(substituted phenyl) Butanoic Acids (**2**) and Their Esters (**3**)

Preparation of 4- oxo-4-(substituted phenyl) butanoic acids (**2**). These Compounds were synthesized according to reports [26,27]. Succinic anhydride (1 equiv, 25 mmol) in 30 mL CH_2_Cl_2_ reacted with substituted benzenes (1, 25 mmol) under stirring for 0.5 h in ice-water bath, then anhydrous aluminum chloride (37.5 mmol) was added to the CH_2_Cl_2_ solution (Scheme 1). The reaction was then kept on room temperature for 6 h. After the reaction was stopped, icy aqueous hydrochloric acid solution (0.1 mol/L, 10 mL) was drop to resultant solution under stirring for 10 min. Organic phase solution was evaporated to give crude product. The crude product was further purified by crystallization repeatedly with 70% ethanol solution to afford 4-oxo-4-phenylbutanoic acid (**2A**) and4-oxo-4-(p-tolyl)butanoic acid (**2B**) [26], 4-(4-Methoxyphenyl)-4-oxobutanoic acid (**2C**) and 4-(4-Chlorophenyl)-4-oxobutanoic acid (**2D**) [27].

Reagents and conditions: (a) Succinic anhydride, AlCl_3_, CH_2_Cl_2_, 0.5h, 0 °C, then 6h rt; (b) Cyclohexanol, TsOH, CH_2_Cl_2_, 4h, reflux; (c) Hydroxylamine hydrdochloride (methoxylamine hydrochloride, benzylhydroamine hydrochloride), pyridine, CH_2_Cl_2_,12h, reflux.

Preparation of Cyclohexyl 4-(4-substituted phenyl) -4-oxobutanoates (**3**) Compound **2** (**2A**, **2B**, **2C**, **2D**, 15 mmol) reacted with cyclohexanol (30 mmol) in CH_2_Cl_2_ (20 mL) with p-methylbenzenesulfonic acid (TsOH) (2g, 11.6 mmol) under refluxing for 4 h (Scheme 1). After stopping, the resultant solution was washed with icy aqueous sodium bicarbonate solution (3%, 15 mL) for three times, then the organic solution was dried with anhydrous sodium sulfate evaporated to give crude product. This crude product was purified by flash column chromatography eluting with an eluent of ethyl acetate/petroleum ether (1:7 *v*/*v*) to afford **3A**, **3B**, **3C**, and **3D**, respectively.

Cyclohexyl 4-oxo-4-phenylbutanoate (**3A**) Colorless liquid, yield 91.6%. ^1^H NMR (CDCl_3_) δ 7.98 (d, *J* = 7.3 Hz, 2H, H-2, 6), 7.55 (t, *J* = 7.4 Hz, 1H, H-4), 7.45 (t, *J* = 7.8 Hz, 2H, H-3, 5), 4.81–4.75 (m, 1H, H-14), 3.29 (t, *J* = 6.7 Hz, 2H, H-8), 2.74 (t, *J* = 6.6 Hz, 2H, H-9), 1.84 (dd, *J* = 12.6, 4.2 Hz, 2H, H-15), 1.70 (dd, *J* = 8.9, 4.3 Hz, 2H, H-19), 1.52 (dd, *J* = 8.9, 3.9 Hz, 1H, H-17), 1.45–1.32 (m, 4H, H-16,18), 1.24 (dd, *J* = 20.5, 7.5 Hz, 1H, H-17). ^13^C NMR (CDCl_3_) δ 198.23 (C-7), 172.29 (C-10), 136.66 (C-1), 133.14 (C-4), 128.59 (C-2, 6), 128.01 (C-3, 5), 72.90 (C-14), 33.44 (C-8), 31.57 (C-15, 19), 28.69 (C-9), 25.37 (C-17), 23.71 (C-16, 18) [28].

Cyclohexyl 4-oxo-4-(p-tolyl) butanoate (**3B**). Colorless crystal, yield 90.8%. m.p. 40.8–41.6 °C. ^1^H NMR (MeOD) δ 7.90 (d, *J* = 8.2 Hz, 2H, H-2, 6), 7.31 (d, *J* = 8.0 Hz, 2H, H-3, 5), 4.76–4.72 (m, 1H, H-15), 3.32–3.28 (m, 2H, H-8), 2.70–2.68 (m, 2H, H-9), 2.41 (s, 3H, H-14, -phCH_3_), 1.82 (dd, *J* = 11.8, 5.7 Hz, 2H, H-16), 1.71 (dd, *J* = 9.7, 3.3 Hz, 2H, H-20), 1.53 (dd, *J* = 11.0, 8.0 Hz, 1H, H-18), 1.45–1.33 (m, 4H, H-17,19), 1.29 (dd, *J* = 12.8, 9.8 Hz, 1H, H-18). ^13^C NMR (CDCl_3_) δ 197.88 (C-7), 172.41 (C-10), 143.92 (C-1), 134.22 (C-4), 129.27 (C-2, 6), 128.15 (C-3,5), 72.89 (C-15), 33.34 (C-8), 31.59 (C-16, 20), 28.76 (C-9), 25.39 (C-18), 23.73 (C-17, 19), 21.64 (C-14). ESIMS: *m*/*z* 275.1646 [M + H]^+^, 297.1461 [M + Na]^+^, calculated for C_17_H_22_O_3_ (274.36).

Cyclohexyl 4-(4-methoxyphenyl)-4-oxobutanoate (**3C**). Colorless crystal, yield 90.2%. m.p. 43.7–44.6 °C. ^1^H NMR (MeOD) δ 7.99 (d, *J* = 9.0 Hz, 2H, H-2, 6), 7.01 (d, *J* = 9.0 Hz, 2H, H-3, 5), 4.77–4.70 (m, 1H, H-16), 3.88 (s, 3H, H-15, OCH_3_), 3.30–3.26 (m, 2H, H-8), 2.70–2.67 (m, 2H, H-9), 1.82 (dd, *J* = 12.0, 5.7 Hz, 2H, H-17), 1.72 (dd, *J* = 8.0, 4.9 Hz, 2H, H-21), 1.56–1.50 (m, 1H, H-19), 1.44 (dd, *J* = 14.4, 8.0 Hz, 2H, H-18), 1.38 (dd, *J* = 9.9, 6.7 Hz, 2H, H-20), 1.32–1.26 (m, 1H, H-19). ^13^C NMR (151 MHz, MeOD) δ 197.60 (C-7), 172.77 (C-10), 163.91 (C-4), 130.04 (C-2, 6), 129.55 (C-1), 113.49 (C-3, 5), 72.64 (C-16), 54.68 (C-15), 32.61 (C-8), 31.14 (C-17, 21), 28.29 (C-9), 25.09 (C-19), 23.25 (C-18, 20). ESIMS: *m*/*z* 291.1593 [M + H]^+^, 313.1413 [M + Na]^+^, calculated for C_17_H_22_O_4_ (290.36).

Cyclohexyl 4-(4-chlorophenyl)-4-oxobutanoate (**3D**). Colorless crystal, yield 89.5%. m.p. 67.5–68.7 °C. ^1^H NMR (CDCl_3_) δ 7.92 (d, *J* = 8.7 Hz, 2H, H-2, 6), 7.44 (d, *J* = 8.6 Hz, 2H, H-3, 5), 4.80-4.75 (m, 1H, H-14), 3.26 (t, *J* = 6.6 Hz, 2H, H-8), 2.74 (t, *J* = 6.6 Hz, 2H, H-9), 1.84 (dd, *J* = 11.4, 5.4 Hz, 2H, H-19), 1.71 (dd, *J* = 8.8, 4.4 Hz, 2H, H-15), 1.55-1.51 (m, 1H, H-17), 1.44–1.32 (m, 4H, H-16,18), 1.26 (t, *J* = 11.6 Hz, 1H, H-17). ^13^C NMR (CDCl_3_) δ 197.04 (C-7), 172.16 (C-10), 139.58 (C-4), 134.99 (C-1), 129.45 (C-2, 6), 128.91 (C-3, 5), 73.01 (C-14), 33.40 (C-8), 31.57 (C-15, 19), 28.61 (C-9), 25.36 (C-17), 23.72 (C-16, c18). ESIMS: *m*/*z* 295.1102 [M + H]^+^, 317.0922 [M + Na]^+^, calculated for C_16_H_19_ClO_3_ (294.78).

#### 3.1.2. Preparation of Oximes and Their Ethers

Compound 3 (**3A**, **3B**, **3C**, and **3D**, 3.84mmol) and substituted hydroxylamine hydrochloride (7.7 mmol) in CH_2_Cl_2_ (20 mL) with pyridine (2 mL) was refluxed for 12 h (Scheme 1). After stopping, the resultant solution was evaporated to give a residue under vacuum. This residue was dissolved in ethyl acetate (20 mL), then washed with saturated sodium chloride solution (20 mL × 3). The organic phase was dried with anhydrous sodium sulfate, and evaporated to give a crude product. This crude product was purified by flash column chromatography with an eluent of ethyl acetate/petroleum ether (1:5 to 1:13 *v*/*v*) to afford ethers.

Cyclohexyl (E)-4-hydroxyimino-4-phenylbutanoate (**4A-1**). Colorless crystal, yield 88.1%. m.p. 100.8–102.0 °C. ^1^H NMR (CDC_l3_) δ 9.23 (s, 1H, H-21, COOH), 7.63 (d, *J* = 3.7 Hz, 2H, H-2, 6), 7.44–7.39 (m, 3H, H-3, 4, 5), 4.81–4.74 (m, 1H, H-14), 3.18-3.13 (m, 2H, H-8), 2.65-2.60 (m, 2H, H-9), 1.83 (dd, *J* = 8.7, 3.8 Hz, 2H, H-15), 1.72 (dd, *J* = 9.8, 4.0 Hz, 2H, H-19), 1.57–1.52 (m, 1H, H-17), 1.38 (dd, *J* = 22.6, 12.5 Hz, 4H, H-16, 18), 1.26 (dd, *J* = 16.3, 6.4 Hz, 1H, H-17). ^13^C NMR (CDCl_3_) δ 172.15 (C-10), 158.24 (C-7), 135.21 (C-1), 129.42 (C-4), 128.67 (C-2, 6), 126.36 (C-3, 5), 73.04 (C-14), 31.56 (C-15, 19), 31.00 (C-9), 25.36 (C-8), 23.73 (C-17), 22.03 (C-16, 18). ESIMS: *m*/*z* 276.1601 [M + H]^+^, 298.1420 [M + Na]^+^, calculated for C_16_H_21_NO_3_ (275.35) [29].

Cyclohexyl (E)-4-methoxyimino-4-phenylbutanoate (**4A-2**). Colorless liquid, yield 81.5%. ^1^H NMR (CDCl_3_) δ 7.64 (d, *J* = 1.9 Hz, 2H, H-2, 6), 7.40–7.34 (m, 3H, H-3, 4, 5), 4.78–4.74 (m, 1H, H-14), 4.00 (s, 3H, H-21), 3.08–3.04 (m, 2H, H-8), 2.57–2.53 (m, 2H, H-9), 1.83 (dd, *J* = 8.9, 3.7 Hz, 2H, H-15), 1.72 (dd, *J* = 9.7, 3.9 Hz, 2H, H-19), 1.54 (dd, *J* = 10.1, 2.5 Hz, 1H, H-17), 1.38 (dd, *J* = 22.0, 12.3 Hz, 4H, H-16,18), 1.27 (dd, *J* = 13.6, 5.8 Hz, 1H, H-17). ^13^C NMR (CDCl_3_) δ 172.03 (C-10), 156.98 (C-7), 135.27 (C-4), 129.19 (C-1), 128.53 (C-2, 6), 126.33 (C-3, 5), 72.90 (C-14), 62.01 (C-21), 31.57 (C-15, 19), 31.20 (C-9), 25.38 (C-8), 23.72 (C-17), 22.38 (C-16, 18). ESIMS: *m*/*z* 290.1748 [M + H]^+^, 312.1567 [M + Na]^+^, calculated for C_17_H_23_NO_3_ (289.38).

Cyclohexyl (E)-4-benzyloxyimino-4-phenylbutanoate (**4A-3**). Pale yellow liquid, yield 79.5%. ^1^H NMR CDCl_3_) δ 7.69 (d, *J* = 3.3 Hz, 2H, H-2, 6), 7.46 (d, *J* = 7.1 Hz, 2H, H-3, 5), 7.43–7.38 (m, 5H, H-23,24,25,26,27), 7.35 (t, *J* = 7.3 Hz, 1H, H-4), 5.29 (s, 2H, H-21), 4.78 (dt, *J* = 8.9, 4.6 Hz, 1H, H-14), 3.16–3.12 (m, 2H, H-8), 2.62–2.58 (m, 2H, H-9), 1.84 (dd, *J* = 9.7, 3.6 Hz, 2H, H-15), 1.75–1.72 (m, 2H, H-19), 1.57 (dd, *J* = 7.9, 4.9 Hz, 1H, H-17), 1.44–1.35 (m, 4H, H-16, 18), 1.30 (dd, *J* = 16.6, 6.5 Hz, 1H, H-17). ^13^C NMR (CDCl_3_) δ 172.09 (C-10), 157.36 (C-7), 138.01 (C-22), 135.29 (C-1), 129.27 (C-4), 128.55 (C-24, 26), 128.41 (C-2, 6), 128.11 (C-3, 5), 127.81 (C-25), 126.42 (C-23, 27), 76.35 (C-21), 72.94 (C-14), 31.59 (C-19), 31.25 (C-9), 25.41 (C-8), 23.75 (C-17), 22.57 (C-16, 18). ESIMS: *m*/*z* 366.2070 [M + H]^+^, 388.1886 [M + Na]^+^, calculated for C_23_H_27_NO_3_ (365.47).

Cyclohexyl (E)-4-hydroxyimino-4-(4-methylphenyl)butanoate (**4B-1**). Colorless crystal, yield 88.2%. m.p. 80.1–80.7 °C. ^1^H NMR (MeOD) δ 7.51 (d, *J* = 8.2 Hz, 2H, H-2, 6), 7.20 (d, *J* = 8.0 Hz, 2H, H-3, 5), 4.70 (dt, *J* = 8.7, 4.5 Hz, 1H, H-15), 3.08-3.05 (m, 2H, H-8), 2.54 (t, *J* = 7.9 Hz, 2H, H-9), 2.36 (s, 3H, H-14), 1.80 (d, *J* = 9.3 Hz, 2H, H-16), 1.75–1.71 (m, 2H, H-20), 1.57–1.54 (m, 1H, H-18), 1.39 (dd, *J* = 17.8, 8.7 Hz, 4H, H-17, 19), 1.32 (d, *J* = 11.5 Hz, 1H, H-18). ^13^C NMR (CDCl_3_) δ 172.15 (C-10), 158.23 (C-7), 139.48 (C-1), 132.32 (C-4), 129.35 (C-2, 6), 126.24 (C-3, 5), 72.96 (C-15), 31.57 (C-16, 20), 31.06 (C-9), 25.37 (C-8), 23.73 (C-18), 21.89 (C-17, 19), 21.27 (C-14). ESIMS: *m*/*z* 290.1754 [M + H]^+^, 312.1573 [M + Na]^+^, calculated for C_17_H_23_NO_3_ (289.38).

Cyclohexyl (E)-4-methoxyimino-4-(4-methoxyphenyl)butanoate (**4B-2**). Colorless liquid, yield 81.7%. ^1^H NMR (CDCl_3_) δ 7.55 (d, *J* = 8.2 Hz, 2H, H-2, 6), 7.19 (d, *J* = 8.0 Hz, 2H, H-3, 5), 4.80–4.73 (m, 1H, H-15), 3.99 (s, 3H, H-22), 3.06-3.03 (m, 2H, H-8), 2.56–2.53 (m, 2H, H-9), 2.38 (s, 3H, H-14), 1.83 (dd, *J* = 8.8, 3.8 Hz, 2H, H-20), 1.73 (dd, *J* = 9.0, 4.7 Hz, 2H, H-16), 1.55 (dd, *J* = 9.6, 6.5 Hz, 1H, H-18), 1.39 (dd, *J* = 22.4, 12.4 Hz, 4H, H-17, 19), 1.29–1.24 (m, 1H, H-18). ^13^C NMR (CDCl_3_) δ 172.11 (C-10), 156.95 (C-7), 139.22 (C-4), 132.40 (C-1), 129.24 (C-2, 6), 126.22 (C-3, 5), 72.87 (C-15), 61.93 (C-22), 31.59 (C-16, 20), 31.27 (C-9), 25.39 (C-8), 23.73 (C-18), 22.34 (C-17, 19), 21.25 (C-14). ESIMS: *m*/*z* 326.1726 [M+Na]^+^, calculated for C_18_H_25_NO_3_ (303.40).

Cyclohexyl (E)-4-benzyloxyimino-4-(4-methylphenyl)butanoate (**4B-3**). Pale yellow liquid, yield 79.3%. ^1^H NMR (CDCl_3_) δ 7.37–7.29 (m, 7H, H-2, 6, 24, 25, 26, 27, 28), 7.22 (d, *J* = 7.9 Hz, 2H, H-3, 5), 5.09 (s, 2H, H-22), 4.74 (dt, *J* = 8.9, 4.6 Hz, 1H, H-15), 2.87–2.83 (m, 2H, H-8), 2.57–2.54 (m, 2H, H-9), 2.38 (s, 3H, H-14), 1.84-1.78 (m, 2H, H-16), 1.72 (dd, *J* = 9.4, 3.6 Hz, 2H, H-20), 1.55 (dd, *J* = 6.9, 3.7 Hz, 1H, H-18), 1.41–1.33 (m, 4H, H-17, 19), 1.27 (t, *J* = 7.0 Hz, 1H, H-18). ^13^C NMR (CDCl_3_) δ 172.16 (C-10), 155.68 (C-7), 138.81 (C-4), 138.41 (C-23), 130.80 (C-1), 128.86 (C-3, 5), 128.22 (C-25, 27), 127.89 (C-26), 127.82 (C-24, 28), 127.47 (C-2, 6), 75.89 (C-22), 72.64 (C-15), 31.61 (C-16, 20), 31.45 (C-9), 30.75 (C-8), 25.40 (C-18), 23.76 (C-17, 19), 21.37 (C-14). ESIMS: *m*/*z* 380.2223 [M + H]^+^, 402.2037 [M + Na]^+^, calculated for C_24_H_29_NO_3_ (379.50).

Cyclohexyl (E)-4-hydroxyimino-4-(4-methoxyphenyl)butanoate (**4C-1**). Colorless crystal, yield 89.0%. m.p. 67.7–68.5 °C. ^1^H NMR (CDCl_3_) δ 8.69 (s, 1H, H-23), 7.59 (d, *J* = 8.9 Hz, 2H, H-2, 6), 6.93 (d, *J* = 8.9 Hz, 2H, H-3, 5), 4.79-4.75 (m, 1H, H-16), 3.85 (s, 3H, H-15), 3.13- 3.10 (m, 2H, H-8), 2.62-2.59 (m, 2H, H-9), 1.83 (dd, *J* = 8.8, 4.0 Hz, 2H, H-17), 1.72 (dd, *J* = 8.7, 3.7 Hz, 2H, H-21), 1.54 (dd, *J* = 5.7, 3.2 Hz, 1H, H-19), 1.38 (dd, *J* = 23.1, 11.8 Hz, 4H, H-18, 20), 1.26 (dd, *J* = 13.0, 3.1 Hz, 1H, H-19). ^13^C NMR (CDCl_3_) δ 172.27 (C-10), 160.61 (C-4), 157.68 (C-7), 127.71 (C-2, 6), 127.63 (C-1), 114.05 (C-3, 5), 73.00 (C-16), 55.33 (C-15), 31.57 (C-17, 21), 31.08 (C-9), 25.37 (C-8), 23.73 (C-19), 21.95 (C-18, 20). ESIMS: *m*/*z* 306.1701 [M + H]^+^, 328.1521 [M + Na]^+^, calculated for C_17_H_23_NO_4_ (305.37).

Cyclohexyl (E)-4-methoxyimino-4-(4-methoxyphenyl)butanoate (**4C-2**). Colorless liquid, yield 80.7%. ^1^H NMR (MeOD) δ 7.59 (d, *J* = 8.9 Hz, 2H, H-2, 6), 6.94 (d, *J* = 8.9 Hz, 2H, H-3, 5), 4.72–4.64 (m, 1H, H-16), 3.95 (s, 3H, H-23), 3.83 (s, 3H, H-15), 3.03–2.97 (m, 2H, H-8), 2.52-2.46 (m, 2H, H-9), 1.82-1.76 (m, 2H, H-17), 1.74-1.67 (m, 2H, H-21), 1.55 (dd, *J* = 10.7, 4.6 Hz, 1H, H-19), 1.42-1.33 (m, 4H, H-18,20), 1.29 (dd, *J* = 9.6, 3.0 Hz, 1H, H-19). ^13^C NMR (MeOD) δ 172.28 (C-10), 160.78 (C-4), 156.67 (C-7), 127.48 (C-2, 6), 127.35 (C-1), 113.53 (C-3, 5), 72.89 (C-16), 60.84 (C-23), 54.42 (C-15), 31.16 (C-17, 21), 30.99 (C-9), 25.07 (C-8), 23.34 (C-19), 21.81 (C-18, 20). ESIMS: *m*/*z* 320.1845 [M + H]^+^, 342.1661 [M + Na]^+^, calculated for C_18_H_25_NO_4_ (319.40).

Cyclohexyl (E)-4-benzyloxyimino-4-(4-methoxyphenyl)butanoate (**4C-3**). Pale yellow liquid, yield 76.9%. ^1^H NMR (MeOD) δ 7.58 (d, *J* = 8.9 Hz, 2H, H-2, 6), 7.41 (d, *J* = 7.1 Hz, 2H, H-25, 29), 7.36 (t, *J* = 7.5 Hz, 2H, H-26, 28), 7.30 (t, *J* = 7.3 Hz, 1H, H-27), 6.92 (d, *J* = 8.9 Hz, 2H, H-3, 5), 5.19 (s, 2H, H-23), 4.66 (dt, *J* = 8.7, 4.5 Hz, 1H, H-16), 3.81 (s, 3H, H-15), 3.06-3.03 (m, 2H, H-8), 2.52-2.49 (m, 2H, H-9), 1.75 (dd, *J* = 13.6, 6.0 Hz, 2H, H-17), 1.68 (dd, *J* = 9.2, 4.9 Hz, 2H, H-21), 1.55–1.50 (m, 1H, H-19), 1.39–1.32 (m, 4H, H-18,20), 1.30–1.24 (m, 1H, H-19). ^13^C NMR (MeOD) δ 172.35 (C-10), 160.82 (C-4), 157.11 (C-7), 138.08 (C-24), 127.99 (C-26, 28), 127.79 (C-2, 6), 127.52 (C-27), 127.43 (C-25, 29), 127.38 (C-1), 113.50 (C-3, 5), 75.75 (C-23), 72.87 (C-16), 54.39 (C-15), 31.09 (C-17, 21), 30.99 (C-9), 25.04 (C-8), 23.28 (C-19), 22.02 (C-18, 20). ESIMS: *m*/*z* 396.2179 [M + H]^+^, 418.1994 [M +Na]^+^, calculated for C_24_H_29_NO_4_ (395.50).

Cyclohexyl (E)-4-hydroxyimino-4-(4-chlorophenyl)butanimidate (**4D-1**). Colorless crystal, yield 86.3%. m.p. 92.9-93.6 °C. ^1^H NMR (CDCl_3_) δ 9.12 (s, 1H, H-21), 7.57 (d, *J* = 8.6 Hz, 2H, H-2, 6), 7.37 (d, *J* = 8.6 Hz, 2H, H-3, 5), 4.76 (dt, *J* = 8.9, 4.6 Hz, 1H, H-14), 3.12-3.09 (m, 2H, H-8), 2.62–2.59 (m, 2H, H-9), 1.84–1.80 (m, 2H, H-15), 1.71 (dd, *J* = 9.5, 3.7 Hz, 2H, H-19), 1.54 (dd, *J* = 9.7, 3.2 Hz, 1H, H-17), 1.37 (dd, *J* = 20.0, 11.6 Hz, 4H, H-16, 18), 1.28–1.23 (m, 1H, H-17). ^13^C NMR (CDCl_3_) δ 172.06 (C-10), 157.39 (C-7), 135.48 (C-4), 133.65 (C-1), 128.87 (C-3, 5), 127.68 (C-2, 6), 73.19 (C-14), 31.56 (C-15,19), 30.91 (C-9), 25.34 (C-8), 23.73 (C-17), 21.89 (C-16, 18). ESIMS: *m*/*z* 332.1024 [M + Na]^+^, calculated for C_16_H_20_ClNO_3_ (309.79).

Cyclohexyl (E)-4 -methoxyimino-4-(4-chlorophenyl)butanoate (**4D-2**). Colorless liquid, yield 75.5%. ^1^H NMR (CDCl_3_) δ 7.59 (d, *J* = 8.6 Hz, 2H, H-2, 6), 7.34 (d, *J* = 8.6 Hz, 2H, H-3, 5), 4.75 (dt, *J* = 9.0, 4.6 Hz, 1H, H-14), 3.99 (s, 3H, H-21), 3.03-3.00 (m, 2H, H-8), 2.53 (dd, *J* = 8.6, 7.5 Hz, 2H, H-9), 1.83-1.79 (m, 2H, H-19), 1.71 (dd, *J* = 9.3, 3.5 Hz, 2H, H-15), 1.56-1.51 (m, 1H, H-17), 1.37 (dd, *J* = 18.8, 10.5 Hz, 4H, H-16,18), 1.28-1.22 (m, 1H, H-17). ^13^C NMR (CDCl_3_) δ 171.90 (C-10), 155.87 (C-7), 135.19 (C-4), 133.71 (C-1), 128.70 (C-2, 6), 127.61 (C-3, 5), 72.99 (C-14), 62.13 (C-21), 31.57 (C-15, 19), 31.10 (C-9), 25.35 (C-8), 23.72 (C-17), 22.16 (C-16, 18). ESIMS: *m*/*z* 324.1370 [M + H]^+^, 346.1187 [M + Na]^+^, calculated for C_17_H_22_ClNO_3_ (323.82).

Cyclohexyl (E)-4-benzyloxyimino-4-(4-chlorophenyl)butanoate (**4D-3**). Pale yellow liquid, yield 70.8%. ^1^H NMR (CDCl_3_) δ 7.60 (d, *J* = 8.4 Hz, 2H, H-2, 6), 7.42 (ddd, *J* = 19.1, 15.0, 6.3 Hz, 10H, H-25, 26, 27, 28, 29, 17, 18, 19, 20, 21), 7.22 (d, *J* = 7.8 Hz, 2H, H-3, 5), 5.29 (s, 2H, H-15), 5.14 (s, 2H, H-23), 3.19-3.16 (m, 2H, H-8), 2.71–2.65 (m, 2H, H-9), 2.41 (s, 3H, H-14). ^13^C NMR (CDCl_3_) δ 172.52 (C-10), 157.05 (C-7), 139.37(C-4), 138.04 (C-24), 135.91 (C-16), 132.32 (C-4),129.30 (C-2, 6), 129.29 (C-26, 28), 128.60 (C-18, 20), 128.42 (C-27), 128.27 (C-19), 128.17 (C-25, 29), 127.81 (C-17, 21), 126.30 (C2, 6), 76.33 (C-23), 66.47 (C-15), 30.95(C-9), 22.45 (C-8), 21.32 (C-14). ESIMS: *m*/*z* 388.1918 [M + H]^+^, 410.1739 [M + Na]^+^, calculated for C_25_H_25_NO_3_ (387.48).

#### 3.1.3. Evaluation of Compounds for Biological Activity in vitro

##### Cytotoxicity of Compounds In Vitro

These synthesized compounds were evaluated for anti-HBV activity with referring to reported method [25,30]. The HepG2.2.15 cell, a human hepatoblastoma cell line stably transfected with cloned HBV DNA, was obtained from Chinese Academy of Medical Sciences (Beijing, China), and was cultured in Dulbecco’s Modified Eagle’s Medium (DMEM) supplemented with 10% fetal bovine serum and 1.5 g/L of sodium bicarbonate, 10 mL/L of streptomycin, 10 mL/L of penicillin, and 200 mg/L of G418 at 37 °C in a 5% CO_2_ atmosphere with 95–98% humidity. Then, HepG2.2.15 cell was plated at a density of 1 × 10^5^ cells/mL in 96-well plates and incubated at 37 °C for 24 h and then treated with variant concentration of compounds in 300, 150, 75, and 35.5 μM/mL; every three days, in a nine day period, C (OD) values were recorded at 450 nm. After nine days, the percent of cell death was measured using the method of MTT assay after drug treatment. Cytotoxicity of these compounds was expressed as the concentration of compound required to kill 50% (CC_50_) of the HepG 2.2.15 cells.

Method for the HBsAg and HBeAg Inhibition Assays. The levels of HBV surface antigen (HBsAg) and HBV e antigen (HBeAg) in the supernatant of the HepG2.2.15 cells were determined by the enzyme-linked immunosorbent assay (ELISA) according to the manufacturer’s protocol (Shanghai Kehua Biotech Co., Ltd., Shanghai, China). The synthesized compounds were expressed as the concentration that achieved 50% inhibition (IC_50_) to the secretion of HBsAg and HBeAg. The selectivity index (SI) was determined as the ratio of CC_50_ to IC_50_ which a major pharmaceutical parameter of estimates possible future clinical application.

### 3.2. Statistical Analysis

SPSS 20.0 and MICROSOFT EXCEL 2010 were used for the statistical analysis. All data were expressed as the mean ± standard error of mean (S.E.M.). The differences among the groups were analyzed by one-way analysis of variance (ANOVA) with a Tukey post hoc test when comparing multiple groups. P values lower than 0.05 were regarded as statistically significant.

## 4. Conclusions

A series of cyclohexyl (E)-4-(alkyl imino)-4-(4-substituted-phenyl) butanoates possessed obvious and significant anti-HBV activities with SI_HBsAg_ values from 0.89 to 13.41 and SI_HBeAg_ values from 1.06 to 17.34 compared to control lamivudine in SI_HBsAg_ values 2.20 and SI_HBeAg_ values 2.07. Results also revealed that oximes were weaker in inhibiting HBV, and more cytotoxic than their relative ethers; meanwhile, methyl ethers were the most active in inhibiting HBV among compounds in every group. Compound **4B-2** among these compounds showed the best inhibition of HBsAg and HBeAg secretion.

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
