# Peer review of "Design, Synthesis, and Bioactive Screen In Vitro of Cyclohexyl (*E*)-4-(Hydroxyimino)-4-Phenylbutanoates and Their Ethers for Anti-Hepatitis B Virus Agents"

_molecules, 2019, doi:10.3390/molecules24112063_

Round 1

Reviewer 1 Report

the publication submitted presents the synthesis of 12 compounds and their evaluation for thier anti HBV activity. These compounds are all cyclohyxyl-hydroxyimino-phenylbutanoate derivatives obtained by a simple and effective synthesis involving a reaction of Friedel and Crafts, esterification and formation of oximes. The synthesized compounds are well characterized. Compounds 2 do not have Mass, perhaps they are already known (reference) but this is not specified.

The experimental part should be reviewed for the description of the coupling constant , according to the guidelines.

The results of the antiviral activity are well presented  and the cacul of the SI informs on the scope of the results. The laters are then explained by an informative docking study.

The results merit publication but I encourage the authors to take more care in writing their future manuscripts (compound numbering in bold, italic, false references ...)

Author Response

The manuscript has been revised according to reviewer’s comment.

Compound’s numbers have been changed to bold style, and coupling constant J and in vitro were changed to italic style J and in vitro.

Other spellings and grammar mistakes has been revised and changed.

As known compounds, detail data of compounds group 2 synthesized and elucidated according to reported articles has been deleted and made the article short.

Reviewer 2 Report

The paper describes the synthesis of a series of gamma-Oximoesters, characterization and testing against hepatitis B cell lines. Of course, the evaluation of new compound lead structures and the optimization of these compounds against HBV is highly relevant and these studies should be supported (and the results published). So, I do recommend publication; the authors should think about some changes in the paper: a) in the synthetic part, the first step is really trivial and I suppose, most compounds are known. So, a major part could be deleted here and one or two characteristic examples can be included. All new compounds - of course - have to appear here;

b) the question of reference and standard should be discussed in more detail: the reference compound is the well-known and multiactive drug lamivudine, a cytidine nucleoside analog. It would be good to give the structure of this compound and also comment on the relatively low activity of this compound and the actual problems with respect to HIV and HBV resistances. This would put the results obtained by the authots into a better surroundinig.

Author Response

Answers: 

 (A) As known compounds, detail data of compounds group 2 synthesized and elucidated according to reported articles was deleted and made the article short.

As known compounds synthesized according to reports, physical property and NMR data are sufficient to elucidate their structures. And more important is as intermediates they were used to synthesized and give correct terminal products.

(B) We thought lamivudine is a famous nucleoside analog, so we did not provide its structure.

Comments about resistance of lamivudine and other nucleoside analogs to HBV had been discussed in introductions.

Reviewer 3 Report

The paper deals with the synthesis, and in vitro screening of a series of  specific molecules Cyclohexyl (E)-4-(Hydroxyimino)-4-Phenylbutanoates and their Ethers as Anti-Hepatitis B Virus Agents. The synthetic work is well described as well as the characterization of the new compounds essentially performed by High field NMR and MS spectroscopy. Most of the studied compounds demonstrated anti-HBV activities. Docking studies were also performed highlighting possible interactions of the series of synthesized compounds with a specific target site.

The paper could be published in Molecules subject to a minor but interesting clarification. All the N-OH (protic) compounds 4A-1, 4B-1, 4C-1 and 4D-1 exhibit  similar characteristics such as weaker anti-HBsAg and anti-HBeAg activities as well as weaker affinities to the selected target protein in docking than those compounds which R=Me and benzyl. According to the NMR Data discussion 1H NMR spectra of the  derivatives showed a singlet at about 8.69~9.23 ppm corresponding to N-O-H proton in compounds 4A-1~4A-3, 4B-2~4B-3, 4C-1~4C-3, 4D-1~4D-3 while only compound 4B-1 was deuterated.  The Authors should better discuss and possibly explain this specific point.

Author Response

Answer:

1H NMR spectra of the derivatives showed that singlets at about 8.69~9.23 ppm were assigned to N-O-H proton in oxime compounds 4A-1, 4C-1 and 4D-1 (Compound 4B-1 was deuterated in CH3OD).